**Data Availability Statement:** All relevant data are within the manuscript and its Supporting Information files.

# Determinants of prelabor rupture of membrane among pregnant women attending governmental hospitals in Jimma zone, Oromia region, Ethiopia: A multi-center case-control study

**Ebrahim Msaye Assefa**[1]*, **Getachew Chane**[2], **Addis Teme**[2], **Tilahun Alemayehu Nigatu**[2]

1 Department of Biomedical Sciences, School of Medicine, College of Medicine and Health Sciences, Wollo University, Dessie, Ethiopia, 2 Department of Biomedical Sciences, Institute of Health, Jimma University, Jimma, Ethiopia

* ebrahimmsaye@gmail.com

## Abstract

### Background

Prelabor rupture of membrane defined as the rupture of fetal membranes before the beginning of uterine contractions, is a common complication of pregnancy and the leading cause of preterm birth. In Ethiopia, the prevalence of prelabor rupture of membrane varied significantly between settings due to variations in risk factors. Besides, there was no study conducted using primary data, particularly in the Jimma zone, Ethiopia. Therefore, this study aimed to identify determinants of prelabor rupture of membrane among pregnant women attending governmental hospitals in the Jimma zone, Oromia region, Ethiopia.

### Methods

An institutional-based unmatched case-control study design was conducted from October 15 to December 15, 2021, at four governmental hospitals. A consecutive sampling technique was used to select 316 participants (79 cases and 237 controls). Women with prelabor rupture of the membrane were confirmed by history, sterile vaginal examination, and ultrasound as cases, and their counterparts as controls. An interviewer-administered questionnaire was used to collect data on maternal (obstetric, medical, behavioral) and fetal-related characteristics. The data were entered into Epi Data version 4.6 and analyzed using SPSS version 25. Descriptive statistics, bi-variable, and multivariable logistic regression were computed. The odds ratio with a 95% confidence level was used, and the significance level was declared at a p-value < 0.05.

**Funding:** The authors received no specific funding for this work.

**Competing interests:** The authors have declared that no competing interests exist.

**Abbreviations:** AGH, Agaro General Hospital; ANC, Antenatal care; AOR, Adjusted odds ratio; COR, Crude odds ratio; GDM, Gestational diabetes mellitus; HMIS, Health Management Information System; JMC, Jimma Medical Center; MUAC, Mid-upper arm circumference; PIH, Pregnancy induced hypertension; PROM, Prelabor rupture of membrane; SCPH, Seka Chekorsa Primary Hospital; SGGH, Shenen Gibe General Hospital; UTI, Urinary tract infection.

## Results

A total of 316 participants (79 cases and 237 controls) were included in this study. Pregnancy-induced hypertension (AOR = 3.06, 95% CI: 1.41–6.64), history of abortion (AOR = 3.67, 95% CI: 1.56–8.65), urinary tract infections (AOR = 2.61, 95% CI: 1.13–6.06), abnormal vaginal discharge (AOR = 2.65, 95% CI: 1.21–5.79), maternal khat chewing (AOR = 3.40, 95% CI: 1.70–6.80), mid-upper arm circumference less than 23 cm (AOR = 2.80, 95% CI: 1.51–5.19), and fetal presentation (breech) (AOR = 2.63, 95% CI: 1.10–6.28) were determinants of prelabor rupture of membrane among pregnant women.

## Conclusion

This study revealed that the aforementioned factors were found to be determinants of prelabor rupture of membrane among pregnant women. Therefore, hospitals should give focus to the early screening, diagnosis, and treatment of pregnancy-induced hypertension, urinary tract infection, and abnormal vaginal discharge to reduce the burden of prelabor rupture of membranes.

## Introduction

The fetal membrane is a thin tissue that surrounds the fetus during pregnancy and is made up of the amnion, which acts as a structural barrier, and the chorion, which safeguards the fetus from the mother's side of the immune system [1]. Prelabor rupture of membranes (PROM) is defined as the rupture of fetal membranes before the onset of true labor (regular and progressive uterine contractions) [2]. It is distinguished by a clinical history of painless, sudden flow out of watery fluid per vagina, as well as amniotic fluid leakage from the cervix, as demonstrated by a sterile speculum vaginal examination [3].

PROM is among the most common complications of pregnancy and a public health issue in the world, including both developed and developing countries [4, 5]. The incidence of PROM is approximately 5% to 10% of all deliveries globally [6]. Every year, it affects 120,000 pregnancies in the United States [7]. The magnitude of PROM varies in different countries, with a prevalence of 18.7% in China [8], 2.2% in India [9], 8.9% in Mexico [10], 3.1% in Brazil [11], 4.91% in Cameroon [4], 5.3% in Egypt [12], 13.8% in Uganda [5], and 23.5% in Ethiopia [13].

PROM is the principal and recognizable cause of preterm births in 40% to 50% of cases [7, 14, 15]. Among all live births, 5–12% are preterm births in the world [16]. Preterm birth is associated with adverse outcomes such as infection, respiratory distress syndrome or birth asphyxia, intraventricular hemorrhage, and bilirubin encephalopathy [17]. According to the World Health Organization (WHO), the most common cause of children's deaths under the age of 5 years is preterm birth [18]. Additional complications associated with PROM that may affect fetal well-being are oligohydramnios, which leads to a greater risk of chorioamnionitis, fetal infection, cord compression, cord prolapse, and fetal distress. PROM may also cause fetal anomalies such as pulmonary hypoplasia and skeletal deformities due to an inadequate amount of amniotic fluid in the fetal sac [19–21].

The risks associated with PROM for the mother include systemic infections (sepsis), an increased rate of indication for a cesarean section, chorioamnionitis (infection of the amniotic sac), infection of the endometrium after delivery of the fetus (postpartum endometritis), retained placenta, abruption placenta, and death [3, 9, 12].

Literature showed that advanced maternal age [22, 23], low-level education [24], low socio-economic status [25], parity [26, 27], pregnancy-induced hypertension (PIH) [28–30], urinary tract infection (UTI) [4, 5, 31, 32], history of cesarean section [33–35], history of abortion [33, 36, 37], smoking [28, 38], and multi-fetal gestation [4, 24, 38] were some of the risk factors contributing to the occurrence of PROM.

Based on the 2016 Ethiopia Demographic and Health Survey (EDHS) report, only about one-fourth (28%) of pregnant women were informed of the vaginal gush of fluid (PROM) as a sign of pregnancy complications during their antenatal care (ANC) follow-up, and information regarding the determinants of PROM in the Ethiopian setup is very limited [39]. Even though some research has been done in Ethiopia, the prevalence of PROM varied significantly between settings, ranging from 1.4% to 23.5% [13, 31, 40], possibly due to variations in determinants from place to place. Most of these studies used cross-sectional study designs and retrospective methods, covering small-scale areas (single-centered) as well as focusing on maternal and perinatal outcomes, and respondents with term pregnancy (above 37 weeks) were excluded.

The accurate cause of PROM remains uncertain. Despite the presence of a few studies using a case-control study design, there are some conflicting or discrepant results on some determinants for PROM such as smoking, wealth index, and PIH [28, 35], which demands further exploration. In addition, some fetal factors like fetal sex, were not assessed. Lastly, there was no study conducted in our study area regarding the determinants of PROM among pregnant women. Therefore, this study aimed to identify the determinants of PROM using a case-control study among pregnant women attending governmental hospitals in the Jimma zone, Oromia region, Ethiopia.

The findings of this study help healthcare providers achieve early identification of mothers who are at risk during ANC and inform pregnant women about the risk factors for PROM to improve the health of mothers and newborns. The study's findings may serve as a source of information for other researchers and policymakers to develop new and effective health strategies for the reduction of PROM and its impact.

## Materials and methods

### Study design, period, and setting

An institutional-based unmatched case-control study design was employed. The study was carried out in the Jimma zone, Oromia Regional State, Southwest Ethiopia, from October 15 to December 15, 2021. We conducted this study in four randomly selected governmental hospitals found in the Jimma zone, namely: Jimma Medical Center (JMC), Shenen Gibe General Hospital (SGGH), Agaro General Hospital (AGH), and Seka Chekorsa Primary Hospital (SCPH). These hospitals provide maternal and child health care services such as family planning, ANC, delivery services, and treatment for obstetric complications. The selected hospitals are staffed by different health professionals, including senior midwives, integrated emergency obstetrics/gynecology and general surgery, general practitioners, and gynecologists/obstetricians who can correctly diagnose PROM.

### Source and study population

All pregnant women with a gestational age of above 28 weeks attending governmental hospitals in the Jimma zone, Oromia region, Ethiopia, were the source population, whereas selected pregnant women with a gestational age greater than 28 weeks who had been diagnosed with PROM by clinicians and admitted to the selected hospitals were cases. The study population for controls was selected pregnant women with a gestational age greater than 28 weeks who

did not have PROM, which has been confirmed by clinicians and attended at the selected hospitals in the Jimma zone, Oromia region, Ethiopia.

**Case definition.**   **Case:** pregnant women with a gestational age greater than 28 weeks who have been diagnosed with PROM using history, sterile speculum vaginal examination, and ultrasound evaluation by physicians [3]. **Control:** pregnant women with a gestational age greater than 28 weeks who did not have PROM, as confirmed by physicians.

**Inclusion and exclusion criteria.**   For cases: pregnant women with a gestational age greater than 28 weeks who had been diagnosed with PROM by clinicians and admitted to labor and maternity wards of each hospital's obstetric department during the study period.

For controls: pregnant women with a gestational age greater than 28 weeks who did not have PROM, which has been confirmed by clinicians, and who attended to labor and maternity wards of each hospital's obstetric department during the study period. Pregnant mothers who were seriously ill, unable to communicate, who had undergone artificial rupture of fetal membranes, and pregnant mothers with intrauterine fetal death were excluded from the study.

## Sample size determination and sampling techniques

The sample size was calculated using Epi Info version 7 statistical software (developed by the Center for Disease Control and Prevention) for an unmatched case-control study design. Based on the results reported from a related study done in Mekelle, Ethiopia [35], the history of abortion, history of PROM, history of cesarean section, and abnormal vaginal discharge were significant predictors of PROM, and those variables were used to estimate the sample size. It was calculated by using the following conditions: the power of the study = 80%, 95% confidence level, the proportion of exposure among cases (p1), the proportion of exposure among controls (p2), and the ratio of cases to controls (r) = 1:3. Later, a 10% non-response rate was added to each calculated sample size to get the final sample. From the above four potential risk factors for PROM, the history of cesarean section gives the largest sample size, which gives a total of 316 study participants (79 cases and 237 controls).

Concerning the selection procedure, first, four hospitals were randomly selected from eight governmental hospitals located in the Jimma zone. Then, the number of cases and controls were proportionally allocated to each hospital based on the number of pregnant women attended at each selected hospital within two months based on the 2020 Health Management Information System report of each hospital. Accordingly, we included (39 cases and 117 controls) from JMC, (18 cases and 54 controls) from SGGH, (14 cases and 42 controls) from AGH, and (8 cases and 24 controls) from SCPH. Finally, the eligible case was selected consecutively, and the three consecutive controls were selected until the determined sample size was met during the data collection period.

## Data collection tool and procedure

Data were collected by face-to-face interviews with the mothers using a semi-structured questionnaire, which was adapted after reviewing different works of literature. The tool generally has four parts involving socio-demographic characteristics, maternal obstetric and medical characteristics, behavioral and nutritional characteristics, and fetal-related characteristics. However, maternal obstetric, medical, and fetal-related data that could not be addressed by interviews, such as gestational age, UTI, abnormal vaginal discharge, anemia, antepartum hemorrhage, gestational diabetes mellitus (GDM), PIH, fetal sex, number of fetuses, fetal presentation, and polyhydramnios, were collected from patient's medical records and charts. The data were gathered in the labor and maternity wards of each selected hospital. Eight data

collectors and four supervisors, senior midwife professionals who can communicate in local languages, were involved in the data collection process.

The diagnosis of PROM was considered when there was a clinical history (complaint) of sudden painless gush (leak out) of fluid from the vagina before the onset of labor and was confirmed by using a sterile speculum vaginal examination for the pooling of amniotic fluid from the posterior vaginal fornix (cervix) and ultrasound evaluation to demonstrate the oligohydramnios [3]. Amniotic fluid is colorless and may contain vernix. If the fluid is not immediately visible, ask the patient to cough and watch the gush of fluid from the vagina for any leaks [41]. This was documented by physicians on the patient's medical card before admission.

The nutritional status of the mother was assessed by measuring the mid-upper arm circumference (MUAC) at the midpoint between the tips of the shoulder and the elbow of the left arm using a standard MUAC tape. Measurements were taken to the nearest 0.1 cm and recorded in the prepared questionnaire.

Fetal sex was determined using ultrasound based on direct visualization of the fetal perineum, particularly the external genitalia [42]. In Ethiopia, an ultrasound is a safe, non-invasive, and relatively accurate method to determine fetal sex. The procedure was done by obstetric ultrasound machines transabdominal after applying gel and using a 3.5 MHz curvilinear transducer on pregnant mothers to determine the sex of the fetus [43, 44]. This was reported by the physicians. Fetal presentation, the number of fetuses, and polyhydramnios were determined by physical examination and ultrasound evaluation. This was also reported and recorded in the patient's medical record by the physicians.

## Data quality assurance

The questionnaire was initially written in English, then translated into local languages (Amharic and Afan Oromo), and retranslated again into English to ensure consistency. A pretest was conducted in the same study area by taking 5% (4 cases and 12 controls) of the total sample size 10 days before the actual data collection, and necessary modifications were made based on the pre-test results. Training was provided for one day at each hospital for data collectors and supervisors regarding the intent of data collection, the content of the questionnaire, data collection techniques, and how to approach study participants. Continuous close supervision was done by supervisors and the principal investigator.

## Data analysis procedure

The data were checked for completeness and coded. The data entry was done in Epi Data version 4.6, and the data from four hospitals were merged and then exported to SPSS (Statistical Package for Social Science) version 25 for analysis [45]. The status of PROM (present or absent) was coded into 1 = cases and 0 = controls. Frequencies, means, standard deviation, and percentages were used for the descriptive analysis of the data. Bi-variable logistic regression was done, and variables with a p-value of less than 0.25 were transported to the multivariable logistic regression model using the backward stepwise (likelihood ratio) method [46]. Model fitness was checked by using the Hosmer-Lemeshow goodness of fit test (p-value = 0.503). Multicollinearity was also checked by using a variance inflation factor (VIF), which was close to one (1.04–1.16). The odds ratio and 95% confidence level were used to evaluate the strength of the association between dependent and independent variables. Finally, predictor variables with a p-value of less than 0.05 in the multivariable logistic regression model were taken as statistically significant risk factors for PROM. The results were presented in the form of text and tables.

 

### Ethical approval and consent to participate

The study was conducted following the principles of the Declaration of Helsinki. An ethical clearance letter was obtained from the Institutional Review Board of the Institute of Health of Jimma University (Ref. No.: IHRPGJ/565/21). Supportive letters were written to the selected hospitals. Official permission was also received from each hospital manager/medical director. Before data collection, informed written consent from each respondent was obtained after a detailed explanation of the main purpose of the study. The confidentiality of information and privacy of the respondents was maintained while doing physical examinations by letting them have a private room. Each respondent was informed that their participation was voluntary and that they also had the right to stop their participation at any time during the study.

## Results

### Socio-demographic characteristics of the study participants

A total of 316 study participants, 79 cases (pregnant women with PROM) and 237 controls (pregnant women without PROM), have participated in this study, making a response rate of 100%. As a result, (39 cases and 117 controls) from JMC, (18 cases and 54 controls) from SGGH, (14 cases and 42 controls) from AGH, and (8 cases and 24 controls) from SCPH were enrolled in this study. The mean (± SD) age of cases and controls was 26.95 (± 5.51) and 26.22 (± 5.34) respectively. Almost four-fifths, 63 (79.7%) of cases and 195 (82.3%) of controls were found in the age group of 20 to 34 years. Twenty-nine (36.7%) of the cases and 76 (32.1%) of the controls were Muslims by religion, while 54 (68.4%) of cases and 158 (66.7%) of controls were Oromo by ethnicity. Likewise, 47 (59.5%) of the cases and 175 (73.8%) of the controls were living in the urban area, and 77 (97.5%) of the cases and 232 (97.9%) of the controls were married. Concerning occupation, 38 (48.1%) cases and 126 (53.2%) controls were housewives. Concerning educational status, 29 (36.7%) of cases and 82 (34.6%) of controls had no formal education. Nearly one-third, 25 (31.6%) of cases and 79 (33.3%) of controls earned 2001–3000 Ethiopian Birr per month (**Table 1**).

### Maternal obstetric and medical characteristics of the study participants

The study showed that more than three-fourths, 61 (77.2%) of the cases and 180 (75.9%) of the controls, were multigravida and 38 (48.1%) of the cases and 103 (43.5%) of the controls were multiparous. Regarding the gestational age of the pregnant women, 47 (59.5%) of the cases and 135 (57%) of the controls were found between 37 and 42 weeks. Sixty-eight (86.1%) of cases and 216 (91.1%) of controls had ANC follow-up for the index pregnancy. Almost one-half, 33 (48.5%) of cases and 129 (59.7%) of controls had 2–3 ANC visits among pregnant mothers who had ANC follow-up. Six (7.6%) cases and 23 (9.7%) controls had antepartum hemorrhage in this pregnancy. Around one-fifth, 17 (21.5%) of cases and 20 (8.4%) of controls develop PIH, while only 1 (1.3%) of cases and 5 (2.1%) of controls develop GDM.

This study revealed that 13 (16.5%) of the cases and 24 (10.1%) of the controls reported at least one previous history of PROM. Similarly, 16 (20.3%) of the cases and 14 (5.9%) of the controls had a prior history of abortion. Of these, 4 (25%) of cases and 4 (28.6%) of controls had a history of multiple abortions. Besides, 16 (20.3%) of cases and 34 (14.3%) of controls had a prior history of cesarean section. More than one-fourth, 21 (26.6%) of cases and 33 (13.9%) of controls, were diagnosed with anemia. Likewise, 15 (19%) of the cases and 19 (8%) of the controls were affected by UTI during the current pregnancy. Nearly one-fifth, 18 (22.8%) of the cases and 23 (9.7%) of the controls, reported abnormal vaginal discharge in this pregnancy (**Table 2**).

 

**Table 1. Socio-demographic characteristics of pregnant women attending governmental hospitals in Jimma zone, Oromia region, Ethiopia, 2021 (n = 316).**

| Variables | Categories | Cases(n = 79) | Controls(n = 237) |
|---|---|---|---|
| | | Frequency (%) | Frequency (%) |
| Age (years) | <20 | 6 (7.6) | 13 (5.5) |
| | 20–34 | 63 (79.7) | 195(82.3) |
| | ≥35 | 10 (12.7) | 29 (12.2) |
| Religion | Muslim | 29 (36.7) | 76 (32.1) |
| | Orthodox | 25 (31.6) | 83 (35) |
| | Protestant | 16 (20.3) | 54 (22.8) |
| | Catholic | 9 (11.4) | 24 (10.1) |
| Ethnicity | Oromo | 54 (68.4) | 158 (66.7) |
| | Amhara | 19 (24.1) | 62 (26.2) |
| | Others* | 6 (7.6) | 17 (7.2) |
| Residence | Urban | 47 (59.5) | 175 (73.8) |
| | Rural | 32 (40.5) | 62 (26.2) |
| Marital status | Married | 77 (97.5) | 232 (97.9) |
| | Others** | 2 (2.5) | 5 (2.1) |
| Occupation | Housewife | 38 (48.1) | 126 (53.2) |
| | Farmer | 12 (15.2) | 31 (13.1) |
| | Merchant | 16 (20.3) | 34 (14.3) |
| | Government employee | 9 (11.4) | 25 (10.5) |
| | Others*** | 4 (5.1) | 21 (8.9) |
| Educational status | No formal education | 29 (36.7) | 82 (34.6) |
| | Primary education | 16 (20.3) | 56 (23.6) |
| | Secondary education | 19 (24.1) | 65 (27.4) |
| | College and above | 15 (19) | 34 (14.3) |
| Average monthly family income (ETB) | ≤1000 | 9 (11.4) | 23 (9.7) |
| | 1001–2000 | 11 (13.9) | 32 (13.5) |
| | 2001–3000 | 25 (31.6) | 79 (33.3) |
| | 3001–4000 | 20 (25.3) | 65 (27.4) |
| | >4000 | 14 (17.7) | 38 (16) |

* = Gurage, Dawro, Kaffa

** = single, widowed and divorced

*** = student, daily laborers, ETB = Ethiopian Birr

## Behavioral and nutritional characteristics of the study participants

Regarding cigarette smoking, only one (1.3%) case and four (1.7%) controls were passive smokers. Eleven (13.9%) cases and 32 (13.5%) controls reported alcohol consumption during the current pregnancy. More than a quarter, 24 (30.4%) of the cases and 28 (11.8%) of the controls, had a history of khat chewing during this pregnancy. Seven (8.9%) of the cases and 18 (7.6%) of the controls had a history of falls or trauma in the current pregnancy. Nearly one-fourth, 19 (24.1%) of the cases and 48 (20.3%) of the controls reported that they had a history of sexual intercourse during the third trimester of the current pregnancy. Based on MUAC measurement of study participants, 34 (43%) cases and 43 (18.1%) controls were measured at less than 23 cm (**Table 3**).

**Table 2. Maternal obstetric and medical characteristics of pregnant women attending governmental hospitals in Jimma zone, Oromia region, Ethiopia, 2021 (n = 316).**

| Variables | Categories | Cases(n = 79) | Controls (237) |
|---|---|---|---|
| | | Frequency (%) | Frequency (%) |
| Gravidity | Primigravida | 18 (22.8) | 57 (24.1) |
| | Multigravida | 61 (77.2) | 180 (75.9) |
| Parity | Nulliparity | 21 (26.6) | 60 (25.3) |
| | Primipara | 20 (25.3) | 74 (31.2) |
| | Multipara | 38 (48.1) | 103 (43.5) |
| Interpregnancy interval (years) | <2 | 15 (24.6) | 53 (29.4) |
| | ≥2 | 46 (75.4) | 127 (70.6) |
| Gestational age (weeks) | <37 | 26 (32.9) | 81 (34.2) |
| | 37–42 | 47 (59.5) | 135 (57) |
| | >42 | 6 (7.6) | 21 (8.9) |
| ANC follow up for current pregnancy | Yes | 68 (86.1) | 216 (91.1) |
| | No | 11 (13.9) | 21 (8.9) |
| Frequency of ANC follow up | 1 visit | 8 (11.8) | 18 (8.3) |
| | 2–3 visits | 33 (48.5) | 129 (59.7) |
| | ≥4 visits | 27 (39.7) | 69 (31.9) |
| Antepartum hemorrhage | Yes | 6 (7.6) | 23 (9.7) |
| | No | 73 (92.4) | 214 (90.3) |
| PIH | Yes | 17 (21.5) | 20 (8.4) |
| | No | 62 (78.5) | 217 (91.6) |
| GDM | Yes | 1 (1.3) | 5 (2.1) |
| | No | 78 (98.7) | 232 (97.9) |
| History of PROM | Yes | 13 (16.5) | 24 (10.1) |
| | No | 66 (83.5) | 213(89.9) |
| Frequency of history of PROM | One time | 9 (69.2) | 19 (79.2) |
| | Two and above | 4 (30.8) | 5 (20.8) |
| History of abortion | Yes | 16 (20.3) | 14 (5.9) |
| | No | 63 (79.7) | 223 (94.1) |
| Number of abortions | One | 12 (75) | 10 (71.4) |
| | Two and above | 4 (25) | 4 (28.6) |
| History of cesarean section | Yes | 16 (20.3) | 34 (14.3) |
| | No | 63 (79.7) | 203 (85.7) |
| Frequency of cesarean section | One time | 11 (68.8) | 28 (82.4) |
| | Two and above | 5 (31.3) | 6 (17.6) |
| Anemia during current pregnancy | Yes | 21 (26.6) | 33 (13.9) |
| | No | 58 (73.4) | 204 (86.1) |
| UTI during current pregnancy | Yes | 15 (19) | 19 (8) |
| | No | 64 (81) | 218 (92) |
| Abnormal vaginal discharge during current pregnancy | Yes | 18(22.8) | 23 (9.7) |
| | No | 61 (77.2) | 214 (90.3) |

## Fetal-related characteristics of the study participants

The study indicated that nearly half, 35 (44.3%) of the cases and 103 (43.5%) of the controls had a male fetus. Regarding fetal presentation, 14 (17.7%) of cases and 15 (6.3%) of controls had breech presentation. In terms of the number of fetuses, only one (1.3%) case and five

**Table 3. Behavioral and nutritional characteristics of pregnant women attending governmental hospitals in Jimma zone, Oromia region, Ethiopia, 2021 (n = 316).**

| Variables | Categories | Cases(n = 79) Frequency (%) | Controls(n = 237) Frequency (%) |
|---|---|---|---|
| Cigarette smoking during current pregnancy | Passive | 1 (1.3) | 4 (1.7) |
| | Former | 0 | 1 (0.4) |
| | Never | 78 (98.7) | 232 (97.9) |
| Consumed alcohol during current pregnancy | Yes | 11 (13.9) | 32 (13.5) |
| | No | 68 (86.1) | 205 (86.5) |
| Khat chewing during current pregnancy | Yes | 24 (30.4) | 28 (11.8) |
| | No | 55 (69.6) | 209 (88.2) |
| History of fall or trauma in pregnancy | Yes | 7 (8.9) | 18 (7.6) |
| | No | 72 (91.1) | 219 (92.4) |
| Third trimester sexual intercourse | Yes | 19 (24.1) | 48 (20.3) |
| | No | 60 (75.9) | 189 (79.7) |
| MUAC (cm) | <23 | 34 (43) | 43 (18.1) |
| | ≥23 | 45 (57) | 194 (81.9) |

(2.1%) controls had more than one fetus. Moreover, 3 (3.8%) cases and 6 (2.5%) controls had been diagnosed with polyhydramnios (**Table 4**).

## Determinants of prelabor rupture of membrane

A bi-variable logistic regression analysis was done for each independent variable to select candidate variables for multi-variable regression. Variables such as residence, PIH, ANC follow-up, history of PROM, history of abortion, history of cesarean section, anemia, UTI, abnormal vaginal discharge, maternal khat chewing, MUAC less than 23 cm, and fetal malpresentation (being breech) had an association with PROM at a p-value of less than 0.25.

Variables that had an association with PROM in the bi-variable analysis (p-value<0.25), were transferred into a multi-variable logistic regression model using the backward stepwise method, and analysis was done after adjusting for covariates. The model was fit because there was no collinearity between variables, and the Hosmer-Lemeshow test had a p-value of 0.503. It was revealed that PIH, history of abortion, UTI, abnormal vaginal discharge, maternal khat chewing, MUAC less than 23 cm, and fetal malpresentation (being breech) were statistically significant risk factors for PROM (p-value<0.05).

**Table 4. Fetal-related characteristics of pregnant women attending governmental hospitals in Jimma zone, Oromia region, Ethiopia, 2021 (n = 316).**

| Variables | Categories | Cases(n = 79) Frequency (%) | Controls(n = 237) Frequency (%) |
|---|---|---|---|
| Fetal sex | Male | 35 (44.3) | 103 (43.5) |
| | Female | 33 (41.8) | 106 (44.7) |
| | Not sure | 11 (13.9) | 28 (11.8) |
| Fetal presentation | Cephalic | 65 (82.3) | 222 (93.7) |
| | Breech | 14 (17.7) | 15 (6.3) |
| Number of fetuses | Single | 78 (98.7) | 232 (97.9) |
| | Twin | 1 (1.3) | 5 (2.1) |
| Diagnosed polyhydramnios | Yes | 3 (3.8) | 6 (2.5) |
| | No | 76 (96.2) | 231 (97.5) |

In this study, PIH was identified as one of the determinants of PROM. The odds of developing PROM among women who had PIH were 3.06 times (AOR = 3.06, 95% CI: 1.41–6.64) higher than their counterparts. Besides, a history of abortion was found to raise the likelihood of developing PROM. Study participants who had a prior history of abortion were 3.67 times (AOR = 3.67, 95% CI: 1.56–8.65) more likely to develop PROM than their counterparts.

Study participants who had UTIs during this pregnancy showed a significant association with PROM. The odds of developing PROM were 2.61 times (AOR = 2.61, 95% CI: 1.13–6.06) higher among those who had UTIs compared to their counterparts. Similarly, abnormal vaginal discharge was found to be a determinant of PROM. Pregnant women who had abnormal vaginal discharge were 2.65 times (AOR = 2.65, 95% CI: 1.21–5.79) more likely to develop PROM compared with those who had no abnormal vaginal discharge.

The current study found that maternal khat chewing was significantly associated with PROM. Pregnant mothers who chewed khat had 3.4-fold (AOR = 3.40, 95% CI: 1.70–6.80) higher odds of developing PROM compared to the odds of mothers who didn't chew khat. Based on this study, MUAC was identified as a determinant of PROM. Study participants with MUAC measuring less than 23 cm were 2.8 times (AOR = 2.80, 95% CI: 1.51–5.19) more likely to develop PROM than participants with MUAC measuring greater than or equal to 23 cm.

The study indicated that fetal malpresentation was also identified as a determinant of PROM. The odds of developing PROM were 2.63 times (AOR = 2.63, 95% CI: 1.10–6.28) higher for pregnant women with breech presentation of the fetus compared to mothers with cephalic presentation of the fetus (**Table 5**).

## Discussion

Prelabor rupture of membrane (PROM) is one of the most common public health issues globally, including Ethiopia. Its significant impact extended from maternal and perinatal morbidity and mortality to economic impact as a result of patient length of stay and hospital costs such as drug-related and health professional expenses [3, 47]. Prediction and prevention of PROM among pregnant women are key measures to reduce its sequelae. Therefore, early identification of modifiable or treatable risk factors in the local context might help towards the development of evidence-based prevention strategies and appropriate interventions.

This study aimed to identify the determinants of PROM among pregnant women attending governmental hospitals in the Jimma zone, Oromia region, Ethiopia. As a result, PIH, history of abortion, UTI, abnormal vaginal discharge, maternal khat chewing, MUAC, and fetal malpresentation (being breech) were independent predictors of PROM.

In this study, PIH was identified as a determinant of PROM. Pregnant women who had PIH had higher odds of developing PROM than their counterparts. This finding is in agreement with the studies carried out in Southern Ethiopia [28], Uganda [29] and China [30]. It is evident that in PIH, the initiating event is an abnormal or shallow cytotrophoblast invasion of spiral arterioles with insufficient uteroplacental blood flow. This results in an ischemic placenta, which leads to vascular endothelial cell activation by increasing the release of inflammatory cytokines or cell mediators that cause inflammation in the body [48, 49]. As a result, the fetal membranes might be weakened and easily ruptured. Therefore, healthcare providers should emphasize early screening and timely treatment of PIH during ANC follow-up.

A prior history of abortion was another predictor of PROM identified in the current study. In agreement with this result, different studies conducted in Mekelle, Southern and Nekemte, Ethiopia [33–35], Uganda [5], Egypt [50], Iran [36], and China [37] reported that a prior history of abortion was associated with the incidence of PROM. The possible reason might be the risk of intraamniotic infection developing from latent upper genital tract infections in a

**Table 5. Bivariable and multivariable logistic regression analysis for determinants of prelabor rupture of membrane among pregnant women attending governmental hospitals in Jimma zone, Oromia region, Ethiopia, 2021 (n = 316).**

| Variables | Category | Status of PROM | | COR (95% CI) | AOR (95%CI) | p-value |
|---|---|---|---|---|---|---|
| | | Cases: N (%) | Control: N (%) | | | |
| Residence | Urban | 47 (59.5) | 175 (73.8) | 1 | 1 | |
| | Rural | 32 (40.5) | 62 (26.2) | 1.92 (1.13–3.28) | 1.51 (0.80–2.87) | 0.20 |
| ANC follow up | Yes | 68 (86.1) | 216 (91.1) | 1 | 1 | |
| | No | 11 (13.9) | 21 (8.9) | 1.66 (0.76–3.63) | 1.56 (0.62–3.90) | 0.35 |
| PIH | Yes | 17 (21.5) | 20 (8.4) | 2.98 (1.47–6.02) | **3.06** (1.41–6.64) | **0.005*** |
| | No | 62 (78.5) | 217 (91.6) | 1 | 1 | |
| History of PROM | Yes | 13 (16.5) | 24 (10.1) | 1.75 (0.84–3.63) | 1.56 (0.68–3.59) | 0.29 |
| | No | 66 (83.5) | 213 (89.9) | 1 | 1 | |
| History of abortion | Yes | 16 (20.3) | 14 (5.9) | 4.05 (1.87–8.74) | **3.67** (1.56–8.65) | **0.003*** |
| | No | 63 (79.7) | 223 (94.1) | 1 | 1 | |
| History of cesarean section | Yes | 16 (20.3) | 34 (14.3) | 1.52 (0.79–2.93) | 1.87 (0.88–3.97) | 0.10 |
| | No | 63 (79.7) | 203 (85.7) | 1 | 1 | |
| Anemia | Yes | 21 (26.6) | 33 (13.9) | 2.24 (1.20–4.16) | 1.23 (0.56–2.69) | 0.61 |
| | No | 58 (73.4) | 204 (86.1) | 1 | 1 | |
| UTI | Yes | 15 (19) | 19 (8) | 2.69 (1.29–5.59) | **2.61** (1.13–6.06) | **0.025*** |
| | No | 64 (81) | 218 (92) | 1 | 1 | |
| Abnormal vaginal discharge | Yes | 18 (22.8) | 23 (9.7) | 2.75 (1.39–5.42) | **2.65** (1.21–5.79) | **0.015*** |
| | No | 61 (77.2) | 214 (90.3) | 1 | 1 | |
| Maternal khat chewing | Yes | 24 (30.4) | 28 (11.8) | 3.26 (1.75–6.06) | **3.40** (1.70–6.80) | **<0.001*** |
| | No | 55 (69.6) | 209 (88.2) | 1 | 1 | |
| MUAC (cm) | <23 | 34 (43) | 43 (18.1) | 3.41 (1.96–5.94) | **2.80** (1.51–5.19) | **0.001*** |
| | ≥23 | 45 (57) | 194 (81.9) | 1 | 1 | |
| Fetal presentation | Cephalic | 65 (82.3) | 222 (93.7) | 1 | 1 | |
| | Breech | 14 (17.7) | 15 (6.3) | 3.19 (1.46–6.95) | **2.63** (1.10–6.28) | **0.029*** |

COR = crude odds ratio, AOR = adjusted odds ratio, CI = confidence interval

* = variables with statistically significance association at p-value <0.05, 1 = reference category, Hosmer-Lemeshow goodness of fit test (p-value = 0.503)

mother who had a prior history of unsafe abortions without getting proper postabortion care using aseptic techniques [3]. In addition, pregnant mothers with two or more abortions probably had a short cervical length, which raised the incidence of PROM [51]. As a result, women with a history of abortion need to be sensitized by all the attending health professionals on the risk of PROM and advised on the need for close monitoring during their subsequent pregnancies.

In contrast to our study, a study carried out in Thailand [52] found that a previous history of abortion had no statistically significant association with PROM. The discrepancy might be due to the exclusion criteria of the study participants. Unlike in our study, women with a gestational age of less than 37 weeks and malpresentation of fetus were excluded from Thailand's study. Moreover, women who had a previous history of abortion might have received postabortion care with aseptic techniques in Thailand.

Study participants who developed UTI were found to have higher odds of developing PROM. Similarly, different studies conducted in Debre Tabor, Ethiopia [31], Uganda [5], Cameroon [4], and India [38] found that UTI was an independent determinant of PROM. This could be due to the fact that bacterial infections in the urinary tract ascend through the vaginal and cervical canals into the decidua and fetal membrane, which ultimately leads to the

release of prostaglandin and cytokines, thereby causing the cervix to soften and become more susceptible to ascending infections, resulting in PROM. Also, the direct release of bacterial proteolytic enzymes such as proteases, collagenases, or trypsin may cause fetal membrane damage, weakness, and subsequent rupture [53]. Therefore, healthcare providers should screen pregnant mothers for UTI and treat all mothers with UTI during ANC visits.

The present study also revealed that abnormal vaginal discharge was identified as a determinant of PROM. This finding is in agreement with studies conducted in Debre Tabor and Mekelle, Ethiopia [31, 35], Nigeria [54], Cameroon [4], Togo [55] and India [32]. The association might be explained by the presence of various microorganisms in the genital tract that proliferate and invade the amniotic fluid and fetal membranes, leading to PROM. Simultaneously, intra-amniotic infection may increase the activity of the uterus, leading to increased intra-uterine pressure, which in turn puts greater stress on the fetal membranes, resulting in weakness and PROM [2]. Thus, healthcare providers should emphasize early screening, diagnosis, and treatment of abnormal vaginal discharge.

In contrast to our study, a population-based study conducted in Brazil showed that there was no association between genitourinary infections and PROM [11]. This difference might be due to a study done in Brazil that used a larger sample size, a cross-sectional study design, and the exclusion of term pregnant women. Also, it may be attributed to the self-reported and early treatment of these infections by most women in the Brazilian study.

Maternal khat chewing was identified as one of the determinants of PROM in the present study. Pregnant women who chewed khat had higher odds of developing PROM than their counterparts in this pregnancy. Similarly, cross-sectional studies carried out in Eastern Ethiopia [13] and Yemen [56] reported that khat chewing was significantly associated with PROM. This could be because khat by itself was found to cause loss of appetite and decreased absorption of nutrients in the gastrointestinal tract, which in turn decreased the availability of micronutrients essential for the strength of the fetal membrane collagen. The possible justification might also be that a woman who chewed khat had a higher risk of periodontal disease, as reflected by poor oral hygiene, calculus deposits, gingival pigmentation, and tooth loss [57]. Evidence from various studies showed that periodontal disease had an increased risk of the occurrence of PROM [58–60]. The possible reasons might be due to the dissemination of oral pathogens/byproducts and inflammatory mediators via the blood stream into the placenta, fetal circulation, amniotic fluid, and fetal membrane [61]. Therefore, it is important to increase awareness regarding the negative aspects of khat chewing during pregnancy.

The current study indicates that MUAC was found to be another determinant of PROM. Pregnant mothers with MUAC measuring less than 23 cm were positively associated with the occurrence of PROM. This finding is compatible with studies carried out in Debre Tabor and Southern Ethiopia [31, 34]. The possible reason could be that pregnant mothers with MUAC measuring less than 23 cm had a nutritional deficiency, which exposed them to a defective structure of collagen, which in turn raised the likelihood of PROM. Micronutrients such as vitamin C and copper are needed in the formation of collagen to provide integrity to fetal membranes. However, low serum concentrations of copper and ascorbic acid (possibly due to dietary deficiency) may contribute to the abnormal structure of collagen and subsequently lead to weakness and result in PROM [2, 62]. As a result, interventions targeting the nutritional status of pregnant women are needed to reduce the occurrence of PROM.

Fetal malpresentation was also one of the independent predictors of PROM in the current study. Specifically, pregnant women with breech presentation of a fetus had higher odds of developing PROM than mothers with cephalic presentation of a fetus. This finding is supported by studies conducted in Israel [26], India [38], and Indonesia [63]. The possible explanation might be that in breech presentation, especially footling breech (feet first), the fetus's

back is arched upwards and the limbs point downward, which leads to direct contact with the weakest point of the fetal membrane overlying the cervix. In addition, in cases of fetal malpresentation, there is delayed or non-engagement of the presenting part that may increase pressure on the amniotic fluid. This leads to the weakening of the dependent part of the fetal membrane immediately superior to the cervix, thereby increasing the chance of PROM.

## Limitations of the study

This study may be vulnerable to recall bias for some variables related to past events. There may be social desirability bias related to personal and sensitive behaviors like third-trimester sexual intercourse and substance use during pregnancy. This study did not compare the four hospitals concerning determinants of PROM.

## Conclusion

This study revealed that PIH, history of abortion, UTI, abnormal vaginal discharge, maternal khat chewing, MUAC of less than 23 cm, and fetal malpresentation (being breech) were determinants of PROM among pregnant women. Early identification and timely initiation of treatment for PIH, UTI and abnormal vaginal discharge should be done during ANC to reduce the occurrence of PROM. Providing counseling for pregnant mothers about the consequences of abortion on their future pregnancy and increasing awareness regarding the adverse effects of khat chewing should be given attention by healthcare providers. Hospitals should work on improving maternal nutritional status during pregnancy via proper nutrition screening, counseling, and interventions. It is also recommended to provide training for pregnant women to improve awareness and to take preventive measures for the determinants of PROM. Lastly, we recommend other researchers to conduct large-scale studies over a long period. In a future study, it would be interesting to study factors that may have an association with PROM such as periodontal disease and micronutrient deficiency.

## Supporting information

**S1 Checklist. STROBE checklist for the study "Determinants of prelabor rupture of membrane among pregnant women attending governmental hospitals in Jimma zone, Oromia region, Ethiopia: A multi-center case control study".**
(DOCX)

**S1 Dataset.**
(SAV)

## Acknowledgments

We would like to acknowledge the staff members, heads of midwiferies, and administrators of each selected hospital in the Jimma zone for their cooperation during this study. The authors also thank the supervisors, data collectors, and study participants for their vital involvement in the realization of this study.

## Author Contributions

**Conceptualization:** Ebrahim Msaye Assefa, Getachew Chane, Addis Teme, Tilahun Alemayehu Nigatu.

**Data curation:** Ebrahim Msaye Assefa, Getachew Chane, Addis Teme, Tilahun Alemayehu Nigatu.

**Formal analysis:** Ebrahim Msaye Assefa.

**Methodology:** Ebrahim Msaye Assefa.

**Writing – original draft:** Ebrahim Msaye Assefa, Getachew Chane, Addis Teme, Tilahun Alemayehu Nigatu.

**Writing – review & editing:** Ebrahim Msaye Assefa, Getachew Chane, Addis Teme, Tilahun Alemayehu Nigatu.

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
