## [Decision Letter · Decision Letter 0]

14 Aug 2023

PONE-D-23-18994Determinants of prelabor rupture of membrane among pregnant women attending governmental hospitals in Jimma zone, Oromia region, Ethiopia: A multi-center case control studyPLOS ONE

Dear Dr. Assefa,

Thank you for submitting your manuscript to PLOS ONE. After careful consideration, we feel that it has merit but does not fully meet PLOS ONE’s publication criteria as it currently stands. Therefore, we invite you to submit a revised version of the manuscript that addresses the points raised during the review process.

ACADEMIC EDITOR: Please revise the manuscript based on the feedback from reviewers:The manuscript need English revisionPlease follow the PLOs One submission guideline, for example on page 6, the title should be Methods and materials.==============================

We look forward to receiving your revised manuscript.

Kind regards,

Kahsu Gebrekidan

Academic Editor

PLOS ONE

Reviewers' comments:

Reviewer's Responses to Questions

**Comments to the Author**

1. Is the manuscript technically sound, and do the data support the conclusions?

Reviewer #1: Yes

Reviewer #2: Partly

2. Has the statistical analysis been performed appropriately and rigorously? 

Reviewer #1: Yes

Reviewer #2: Yes

3. Have the authors made all data underlying the findings in their manuscript fully available?

Reviewer #1: No

Reviewer #2: No

4. Is the manuscript presented in an intelligible fashion and written in standard English?

Reviewer #1: Yes

Reviewer #2: Yes

5. Review Comments to the Author

Reviewer #1: Dear Author

this is a very interesting paper, nice and easy to read

I wish you used another word to replace the word predictors!, or determinants!

it would be perfect sincenyou only measured associations and not risk ratio

measures of associations only give associations and not causal relations, the predictors and determinants seem more like a causal relations

I would thionk that adding line numbers to the manusript would make it easy to tract the comments

Results: Table 5. is a combination of bivariable and multivariable findings, whay dont you seperate the two tables and report them independently?

STudy Limitations: what did you do to mitigate the limitation of recall bias? please state it clearly as a delimitation

Conclusion:

does your study have something to reccomend for training? it would be nice to say something on practice and training

be specific about what you would like the readers to study next

Reviewer #2: Reviewer reports

Title: Determinants of prelabor rupture of membrane among pregnant women attending governmental hospitals in Jimma zone, Oromia region, Ethiopia: A multi-center case control study

Reviewer: Mesfin Abebe

Thank you for the opportunity to review this manuscript I read the manuscript in detail and the manuscript is well-structured and written in a good manner. However, I raised some major comments and suggestions below. I'm hopeful the comments below can help fill these gaps and improve the quality of the paper.

Major revision

The paper's overall language quality is a little low; there are numerous grammatical errors throughout the entire manuscript. Before the next submission, I recommend that the entire manuscript be revised by a native English speaker.

Abstract:

1. The abstract accurately reflects the details presented in the manuscript's body. However, the use of abbreviations/acronyms in the abstract section is not recommended, and I recommend revising it.

2. You stated as in abstract section “Therefore, hospitals should give focus to the early screening, diagnosis, and treatment of modifiable risk factors to reduce the burden of prelabor rupture of membrane” What are those modifiable risk factors? Better to list those modifiable risk factors.

Background:

1. Your background information is somewhat good, but didn’t show a gap in your study. The author only identified as fetal sex and no prior study conducted in jimma zone as gap. You need to write more than this.

2. You stated as “Therefore, this study aimed to address the identified gaps and examine the determinants of PROM using a case-control study among pregnant women attending governmental hospitals in the Jimma zone, Oromia region, Ethiopia.” Rewrite this sentence as “Therefore, this study aimed to identify the determinants of PROM using a case-control study among pregnant women attending governmental hospitals in the Jimma zone, Oromia region, Ethiopia.”

Methods:

1. Better to remove the word prospective. “An institutional-based unmatched case-control study design was employed” is enough.

2. Why were consecutive sampling technique used to select study participants? Why not used systematic sampling technique?

3. During selection of candidate variable for final model or multivariable logistic regression you only used p value less than 0.25. Why not consider another mechanism? Because you only consider p value you may be loss some important variable. So consider other mechanisms in addition to p value.

Results

1. In table 1; what is your reference for category of average monthly income?

2. What is the difference between Nulliparity and Primipara?

Discussion:

1. The discussion is generally well-organized and incorporates existing literature. There might be more consideration given to the next steps in the research or implementation

2. You stated “recall and social desirability bias as limitation of your study” so what mechanism used to minimize those bias? Because of those bias common in case control study design. Please write down how to minimize those bias beside of limitation of study.

3. In abbreviation section I was see some acronyms. So rewrite as abbreviations and acronyms

6. PLOS authors have the option to publish the peer review history of their article (what does this mean?). If published, this will include your full peer review and any attached files.

Reviewer #1: No

Reviewer #2: **Yes: **Mesfin Abebe

---

## [Author Response · Author response to Decision Letter 0]

3 Oct 2023

thank you for your thorough evaluations and suggestions.

---

## [Decision Letter · Decision Letter 1]

2 Nov 2023

Determinants of prelabor rupture of membrane among pregnant women attending governmental hospitals in Jimma zone, Oromia region, Ethiopia: A multi-center case control study

PONE-D-23-18994R1

Dear Mr. Ebrahim,

We’re pleased to inform you that your manuscript has been judged scientifically suitable for publication and will be formally accepted for publication once it meets all outstanding technical requirements.

Kind regards,

Kahsu Gebrekirstos Gebrekidan

Academic Editor

PLOS ONE

Additional Editor Comments (optional):

Reviewers' comments:

Reviewer's Responses to Questions

**Comments to the Author**

1. If the authors have adequately addressed your comments raised in a previous round of review and you feel that this manuscript is now acceptable for publication, you may indicate that here to bypass the “Comments to the Author” section, enter your conflict of interest statement in the “Confidential to Editor” section, and submit your "Accept" recommendation.

Reviewer #2: All comments have been addressed

2. Is the manuscript technically sound, and do the data support the conclusions?

Reviewer #2: Yes

3. Has the statistical analysis been performed appropriately and rigorously? 

Reviewer #2: Yes

4. Have the authors made all data underlying the findings in their manuscript fully available?

Reviewer #2: Yes

5. Is the manuscript presented in an intelligible fashion and written in standard English?

Reviewer #2: Yes

6. Review Comments to the Author

Reviewer #2: Thank you for the opportunity to review this manuscript again. I thoroughly reviewed the amended document, and it is well-structured and well-written. I have no further remarks because the author has addressed all of my previous comments.

7. PLOS authors have the option to publish the peer review history of their article (what does this mean?). If published, this will include your full peer review and any attached files.

Reviewer #2: **Yes: **Mesfin Abebe

---

## [Editor Report · Acceptance letter]

21 Nov 2023

PONE-D-23-18994R1 

Determinants of prelabor rupture of membrane among pregnant women attending governmental hospitals in Jimma zone, Oromia region, Ethiopia: A multi-center case-control study 

Dear Dr. Assefa:

I'm pleased to inform you that your manuscript has been deemed suitable for publication in PLOS ONE. Congratulations! Your manuscript is now with our production department. 

Kind regards, 

on behalf of

Dr. Kahsu Gebrekidan 

Academic Editor

PLOS ONE